# *Schistosoma japonicum* Tyrosine Hydroxylase is promising targets for immunodiagnosis and immunoprotection of Schistosomiasis japonica

Xianyu Piao[1☯], Jiamei Duan[1☯], Ning Jiang[2,3], Shuai Liu[1], Nan Hou[1‡]*, Qijun Chen[1,2,3‡]*

**1** NHC Key Laboratory of Systems Biology of Pathogens, Institute of Pathogen Biology, Chinese Academy of Medical Sciences & Peking Union Medical College, Beijing, China, **2** Key Laboratory of Livestock Infectious Diseases in Northeast China, Ministry of Education, Key Laboratory of Ruminant Infectious Disease Prevention and Control (East), Ministry of Agriculture and Rural Affairs, College of Animal Science and Veterinary Medicine, Shenyang Agricultural University, Shenyang, China, **3** The Research Unit for Pathogenic Mechanisms of Zoonotic Parasites, Chinese Academy of Medical Sciences, Shenyang, China

☯ These authors contributed equally to this work.
‡ NH and QC contributed equally to this work as corresponding author.
* hounan@ipbcams.ac.cn (NH); qijunchen759@syau.edu.cn (QC)

**Data Availability Statement:** All data are available in the submission and its Supporting Information files.

## Abstract

Identification of promising schistosome antigen targets is crucial for the development of anti-schistosomal strategies. Schistosomes rely on their neuromuscular systems to coordinate important locomotory behaviors. Tyrosine hydroxylase (TH) is critical in the initial rate-limiting step in biosynthesis of catecholamine, the important neuroactive agents, which promote the lengthening of the worm through muscular relaxation and are therefore of great importance to the movement of the organism both within and between its hosts. THs from both *Schistosoma mansoni* and *Schistosoma japonicum* and their enzyme activities have been discovered; however, the role of these proteins during infection have not been explored. Herein, a recombinant protein of the nonconserved fragment of *S. japonicum* TH (SjTH) was produced and the corresponding polyclonal antibody was generated. The expression and antigenicity of SjTH were detected by qRT-PCR, western blotting, immunofluorescence assays, and ELISA. Mice immunized with the recombinant SjTH were challenged with cercariae to evaluate the immunoprotective value of this protein. Our results showed SjTH not only distributed in the head associated with the central nervous system, but also expressed along the tegument and the intestinal intima, which are involved in the movement, coupling and digestion of the parasites and associated with the peripheral nervous system. This protein can effectively stimulate humoral immune responses in mammalian hosts and has high potential as a biomarker for schistosomiasis immunodiagnosis. Furthermore, immunization with recombinant SjTH showed to reduce the worm and egg burden of challenged mice, and to contribute to the systemic balance of the Th1/Th2 responses. Taken together, these results suggest that SjTH is an important pathogenic molecule in *S. japonicum* and may be a possible target for anti-schistosomal approaches.

**Funding:** This work was supported by Chinese Academy of Medical Sciences Innovation Fund for Medical Sciences (CIFMS) [Grant Number 2021-1-I2M-038, received by NH, and 2019-I2M-5-042, received by QC] and the Fundamental Research Funds for the Central University [Grant Numbers 3332021092, received by NH]. The funders had no role in study design, data collection and analysis, decision to publish, or preparation of the manuscript.

**Competing interests:** All authors report no competing interests.

## Author summary

Schistosomiasis is still a disease imposing great threat to human life and which is far from elimination. The identification and function of many schistosome antigens remains limited, which constrains the development of schistosomiasis vaccines and new diagnostics markers. Schistosomes rely on their neuromuscular systems to coordinate important locomotory behaviors. Tyrosine hydroxylase (TH) is critical for normal physiological and neuropathological conditions, which are of great importance to the movement of the parasites both within and between its hosts. THs from both *S. mansoni* and *S. japonicum* and their enzyme activities have been discovered, however, the role of these proteins during infection have not been explored. In this study we show SjTH may be expressed in the head, on the tegument surface, and along the intestinal intima of the parasites, associated with both central nervous system and peripheral nervous system. This protein effectively stimulates the host immune response and exhibited highly encouraging performance for schistosomiasis immunodiagnosis. We also show that immunization with the recombinant SjTH was able to protect mice from a challenge *S. japonicum* infection. Overall, these results indicate the SjTH holds potential value as an immunoprotection target.

## Introduction

Schistosomiasis is estimated to cause 280,000 deaths annually across 78 countries and approximately 3.8 million disability-adjusted life years are credited to this disease [1,2]. The three major schistosome species known to infect humans are *Schistosoma mansoni*, *Schistosoma haematobium*, and *Schistosoma japonicum* [3,4]. Over 90% of schistosomiasis are concentrated in sub-Saharan Africa induced by *S. mansoni* and *S. haematobium*, while Schistosomiasis japonica was largely brought under control and is mainly located in the People's Republic of China, Philippines, and Indonesia [1,2]. Substantial progress has been made in the past decade to eliminate schistosomiasis; nevertheless, the available public health interventions remain inadequate. Identification of promising schistosome antigen targets is crucial for the development of more efficient diagnostic methods, chemotherapeutic drugs, and vaccines.

One area of interest in the development of new anti-schistosomal strategies is the nervous system of the parasites. Catecholamines, such as dopamine, norepinephrine, and epinephrine, are important neuroactive agents in various vertebrates and invertebrates, and are also present in several parasitic helminths, including *S. mansoni* [5,6]. They are inhibitory neurotransmitters that promote the lengthening of the worm through muscular relaxation [7–9], and are therefore of great importance to the movement of the organism both within and between its hosts. Tyrosine hydroxylase (TH) catalyzes the conversion of L-tyrosine into L-dihydroxyphenylalanine, which is rapidly metabolized to produce active catecholamines through a multistep enzymatic pathway. This is the initial rate-limiting step in catecholamine biosynthesis [10–12]. Therefore, regulation of TH activity is a critical step in catecholamine synthesis, and is essential for normal physiology and neuropathological conditions. TH is commonly expressed in the central nervous system (CNS), and is predominantly present in the cytoplasm of cells [13]. However, many studies have reported the present of TH in the brain, gut, retina, sympathetic nervous system, and adrenal medulla of various species [12,14–17], which are associated with the peripheral nervous system (PNS). There are four TH protein isoforms expressed in humans, two in anthropoids and Drosophila, and only one in other species investigated to date, including rats and cows [18,19]. Noteworthy, THs from *S. mansoni* and *S. japonicum*

have been cloned, and their activities have been studied [20,21]. Both recombinant *S. mansoni* TH (SmTH) and *S. japonicum* TH (SjTH) are approximately 54 kDa and have similar structures and catalytic properties to those of the mammalian enzymes. Moreover, TH of schistosoma was found to play an important role in the biosynthesis of catecholamines, similar to the mammalian enzyme, showing the same absolute requirement for a tetrahydrobiopterin cofactor and similar sensitivity to be inhibited by high concentrations of the substrate tyrosine [20,21].

However, there are few studies on schistosome TH, and the only investigations on SmTH and SjTH mainly focused on their enzyme activity features [20,21], while the role of SjTH in the course of infection has not been explored. The present study aimed to provide clues for bridging this knowledge gap; thus, the expression of SjTH in schistosome parasites and SjTH antigenicity in hosts were studied. In addition, mice were immunized with recombinant proteins of the non-conserved fragment of SjTH, and challenge experiments were performed to evaluate the potential of this enzyme as an immunoprotective target for anti-schistosomal approaches.

## Materials and methods

### Ethical statement

All experiments using human samples were performed in accordance with the tenets of the Declaration of Helsinki. Serum samples were donated by patients and healthy volunteers. Written informed consent was obtained from all participants and all information pertaining to the individuals was anonymized. All animal procedures in this study were conducted in accordance with the animal husbandry guidelines of the Chinese Academy of Medical Sciences. Studies on both humans and animals were reviewed and approved by the Ethical Committee and the Experimental Animal Committee of the Chinese Academy of Medical Sciences, with ethical clearance numbers IPB-2016-9 and IPB-2021-6.

### Parasites and animals

Snails (*Oncomelania hupensis*) artificially infected with *S. japonicum* were provided by the Jiangxi Provincial Institute of Parasitic Diseases, Jiangxi, China. The sex of cercariae released by the snails were determined using duplex real-time PCR method [22]. Freshly released cercariae stimulated by light were harvested immediately. Six-week-old pathogen-free male BALB/c mice were percutaneously infected with 20 ± 1 male cercariae and 20 ± 1 female cercaria. New Zealand white rabbits (both obtained from Vital River Laboratory Animal Technology Co., Beijing, China) were percutaneously infected with cercariae (1000 ± 100 per rabbit). Serum samples from the infected animals were collected at 0, 7, 14, 21, 28, 35, 42 and 56 days post infection (dpi). Hepatic schistosomula (14 dpi) and adult worms (42 dpi) were manually isolated by portal perfusion via the vascular system of the infected mice under a light microscope. Eggs were purified from the liver tissues of infected mice 42 dpi by enzyme digestion, as previously described [23].

### Human samples

Twenty patients with schistosomiasis japonica, confirmed by the Kato-Katz method, were recruited from March 2016 to February 2017 in the Hunan province, China. Serum samples were collected before and 3 months after praziquantel treatment. Healthy volunteers (*n* = 20) were recruited from Heilongjiang, a province not endemic for schistosomiasis japonica. Patients with echinococcosis (*n* = 15) were recruited from the Xinjiang Uyghur Autonomous

Region and confirmed by clinical diagnosis, parasitological detection, and medical history records. A summary of patient information is presented in S1 Table.

## Quantitative real-time polymerase chain reaction (qRT-PCR)

The transcriptional expression of *SjTH* (GenBank ID: HQ234745.1) was determined using qRT-PCR. Briefly, total RNA was extracted from parasites using RNeasy Mini Kit (Qiagen, Hilden, Germany). Total RNA (1 μg) was reverse-transcribed into cDNA using the Invitrogen SuperScript III reverse transcriptase kit (Thermo Fisher Scientific, Waltham, MA, USA) according to the manufacturer's instructions. The 26S proteasome non-ATPase regulatory subunit 4 (*PSMD4*; GenBank ID: FN320595) was used as reference gene [24]. Triplicate reactions, detecting the expression of glyceraldehyde-3-phosphate dehydrogenase (*GAPDH*, GenBank ID: FN324551) in standard cDNA (equally mixed cDNA from parasites of the five stages), were arranged for standard controls. The following specific primers were used: *SjTH*, forward: 5′–CACGCTACTAGAGCATGCAA–3′ and reverse: 5′–ACCAGCGACAGGTC GAATAC–3′; *PSMD4*, forward: 5′–CCTCACCAACAATTTCCACATCT–3′ and reverse: 5′–GATCACTTATAGCCTTGCGAACAT–3′; α-Tubulin, forward: 5′-ATGGAACAAG-GATGGTGCTGAG-3′ and reverse: 5′-CAACAAACATGGGTGCGTCT-3′. QRT-PCR was performed in technical triplicates using Brilliant II SYBR Green QPCR Master Mix Kit (Agilent Technologies, Santa Clara, CA, USA) and an Applied Biosystems 7500 Real-time PCR System (Thermo Fisher Scientific), according to the manufacturers' instructions. The data were analyzed using Applied Biosystems 7500 system software version 1.3.1. The gene expression values of the parasite at specific developmental stages were normalized to the standard control (GAPDH in the standard parasite cDNA).

## Recombinant protein production and polyclonal antibody generation

Recombinant proteins of SjTH (rSjTH) were prepared using Invitrogen Gateway Technology with Clonase II (Thermo Fisher Scientific), according to the manufacturer's instructions. Briefly, specific primers comprising 25 bp-attB sites were designed using Primer BLAST (https://www.ncbi.nlm.nih.gov/tools/primer-blast/): forward primer, 5′–GGGGA-CAAGTTTGTACAAAAAAGCAGGCTGACCACCTATTACTGCATCCAA–3′ and reverse primer, 5′–GGGGACCACTTTGTACAAGAAAGCTGGGTACATGGTGATCCTATACCT-GAA–3′. Gene fragments (211–543 bp) of SjTH were amplified from the schistosome cDNA using high-fidelity Phusion DNA polymerase (Finnzymes Oy, Espoo, Finland). The amplified product was cloned into the Gateway entry plasmid pDONR221 (Thermo Fisher Scientific) using the BP recombination reaction and then transferred to the Gateway expression plasmid pDEST17 (Thermo Fisher Scientific) using the LR recombination reaction. *Escherichia coli* Transetta (DE3) cells (TransGen Biotech, Beijing, China) were used to generate recombinant proteins with positive clones. His-tagged fusion proteins were purified using Ni-NTA agarose (Qiagen). Proteins were analyzed by 12% sodium dodecyl sulfate-polyacrylamide gel electrophoresis (SDS-PAGE) and western blotting using monoclonal antibodies against His-tag (Cell Signaling Technology, Danvers, MA, USA). Recombinant *S. japonicum* secreted protein 13 (SjSP-13) was generated as previously described [25]. Rabbit polyclonal antibodies were prepared by Beijing Protein Innovation (Beijing, China) by immunizing New Zealand white rabbits with recombinant proteins.

## Western blotting

The immunogenicity of SjTH was detected by western blotting. Total proteins were separated on 12% SDS-PAGE gels and transferred to polyvinylidene difluoride (PVDF) membranes

(Millipore, Bedford, MA, USA). After blocking with 5% skimmed milk, sera from schistosomiasis japonica patients, *S. japonicum* infected mice, and rabbits were used as primary antibodies (diluted 1:500). Sera from uninfected individuals and animals were used as negative controls. Detection was performed by incubation with IRDye 800 CW conjugated goat anti-human IgG (H+L), goat anti-mouse IgG (H+L), and goat anti-rabbit IgG (H+L) antibodies (all from Li-COR Biosciences, Lincoln, NE, USA).

Parasites, stored at −80°C, were homogenized by grinding in liquid nitrogen, followed by incubation with lysis buffer (8 M urea, 4% CHAPS, 1% dithiothreitol, 1% ethylenediaminetetraacetic acid, 10 mM Tris, and 35 μg/mL phenylmethylsulfonyl fluoride) for 30 min on ice, and then centrifuged at 12,000 rpm for 30 min at 4°C. The protein extracts (10 μg) were separated on 12% SDS-PAGE gels and transferred to PVDF membranes. Rabbit polyclonal antibodies against recombinant SjTH (rSjTH) were used as primary antibodies (dilution 1:1,000), and rabbit IgG was used as a control. Detection was performed by incubation with IRDye 800 CW conjugated goat anti-rabbit IgG (H+L) antibodies, using Odyssey (Li-COR Biosciences).

## Enzyme linked immunosorbent assay (ELISA)

Plates (96-well) were coated with 1 μg/mL rSjTH (100 μL/well) in a coating buffer (Sigma-Aldrich, St. Louis, MO, USA) for more than 10 hours at 4°C. Triplicate reactions of human IgG, mouse IgG, and rabbit IgG were arranged on the plate as positive controls and phosphate-buffered saline (PBS) were used as negative control. Serum samples (100 μL, 1:100 dilution) were added to the wells after blocking with 10% skim milk. Goat anti-mouse, -rabbit, and -human polyvalent immunoglobulin (α-, γ-, and μ-chain specific) conjugated to alkaline phosphatase (Sigma-Aldrich) were used as secondary antibodies (1:10,000 dilution). The reaction was developed using p-nitrophenyl phosphate (Sigma-Aldrich) and stopped with 3 M sodium hydroxide. The optical density (OD) of each well was measured at 405 nm, and the OD values on different plates were weighted by the OD value of the control IgG at 0.1 μg/mL. For the detection of mouse and rabbit sera, the final OD value was calculated as $(OD_{sample}−OD_{blank})/OD_{control\ IgG}$. For the detection of human samples, the cutoff value of the positive test was set at 2.1-times the mean OD value of serum samples of healthy individuals. Sensitivity was defined as true positives/(true positives + false negatives), and specificity as true negatives/(false positives + true negatives).

## Immunofluorescence

The parasites were embedded in OCT compound and serial cryosections (5–7 μm) were obtained. The tissue sections were fixed for 10 min in 4% formaldehyde and rinsed with PBS containing 0.3% Triton X-100, and were then incubated in blocking solution (5% bovine serum albumin in PBS) for 2 h at 25°C, followed by rabbit polyclonal antibodies against rSjTH (2 mg/mL, 1:500 dilution) and IgG from a non-immunized rabbit in blocking solution overnight 4°C. Parasite tissue sections were further incubated with Alexa Fluor 555 donkey anti-rabbit IgG (H+L) and 4', 6-diamidino-2-phenylindole (DAPI, all from Invitrogen, Waltham, MA, USA). Fluorescence was visualized using a TCS SP5 confocal microscope (Leica Microsystems, Wetzlar, Germany).

## *In vivo* immunization and challenge experiments

Mice in each group were subcutaneously injected with 60 μg proteins (rSjTH or SjSP-13) or the same volume of PBS emulsified with complete Freund's adjuvant for the first immunization, and 30 μg protein or PBS emulsified with incomplete Freund's adjuvant every 2 weeks for a total three immunizations. Antibody titers in mouse sera were measured by ELISA, as

described above. After successful immunization, mice were percutaneously challenged with cercariae (40 ± 2 parasites per mice) released from infected *O. hupensis* snails. Basic physical signs of all mice were observed and recorded. Adult worms and eggs were isolated and counted at 42 dpi. The livers of mice 42dpi were removed and serial paraffin sections of 5 μm were obtained. The slides were fixed with 4% paraformaldehyde and stained with hematoxylin and eosin (HE). The histological samples were observed using Nikon ECLIPSE 80i (Nikon, Tokyo, Japan) and the granuloma areas were measured using ImageJ software (https://imagej.nih.gov/ij/index.html). Cytokines in mouse sera were detected using the Mouse TNF-alpha Quantikine ELISA Kit, Mouse IFN-gamma Quantikine ELISA Kit, Mouse IL-4 Quantikine ELISA Kit, and Mouse IL-10 Quantikine ELISA Kit all from R&D Systems, Minneapolis, MN, USA) according to the manufacturer's instructions.

## Statistical analysis

Data were analyzed using Prism 5.0 (GraphPad Software, San Diego, CA, USA) and Microsoft Excel 2010 (Microsoft Corporation, Redmond, WA, USA). The statistical significance of the experimental data was evaluated between two groups using two-tailed paired or nonparametric Student's *t*-tests and among more groups using one-way analysis of variance. Statistical significance was set at $p < 0.05$.

## Results

### SjTH is mainly expressed in hepatic schistosomula and adult worms

Quantitative RT-PCR showed that the transcriptional expression of SjTH was highest in hepatic schistosomula compared with that in the other four parasite stages (Fig 1A). Herein, a polyclonal antibody against SjTH was used to detect SjTH levels. The C-terminus of SjTH is highly conserved, especially its sequence within amino acid positions 190–400 [20]. To ensure the specificity of the antibody, sequence fragments of *SjTH* (211–543 bp) encoding the 71–181 amino acids were amplified to construct the clones. A 14 kDa recombinant protein of partial SjTH was obtained, analyzed by SDS-PAGE and Coomassie brilliant blue staining, and further confirmed by western blotting with an anti-His tag mouse monoclonal antibody (S1 Fig). Western blotting showed that the natural SjTH was approximately 45 kDa, and was mainly expressed in hepatic schistosomula and adults, with very low expression in cercariae and almost no expression in eggs (Fig 1B). Further analysis by immunofluorescence revealed that SjTH was present in the head (Fig 2A), on the tegument surface (Fig 2B, 2C and 2D) and along the intestinal intima (Fig 2B, 2C and 2D) of male adult, female adult and hepatic schistosomulum.

### SjTH-specific antibodies are increased in *S. japonicum*-infected hosts

Antibodies against SjTH in sera derived from *S. japonicum* infected patients, BALB/c mice and New Zealand rabbits 42 dpi were identified by western blotting (S1 Fig). The dynamics of SjTH-specific IgG in mouse and rabbit sera were evaluated at 0, 7, 14, 21, 28, 42, and 56 dpi by ELISA. Overall, the antibody titers in infected mouse sera began to rise at 28 dpi and peaked at 42 dpi (Fig 3A), and those of infected rabbit sera began to rise as early as 14 dpi, and peaked at 28 or 42 dpi (Fig 3B).

ELISAs were further performed to analyze the levels of antibodies against SjTH in human serum samples (Fig 3C). Sera from healthy volunteers were used as negative controls, whereas sera from patients with echinococcosis were used to assess the analysis specificity and cross-reactivity. SjSP-13, which was recently identified as a novel diagnostic protein for

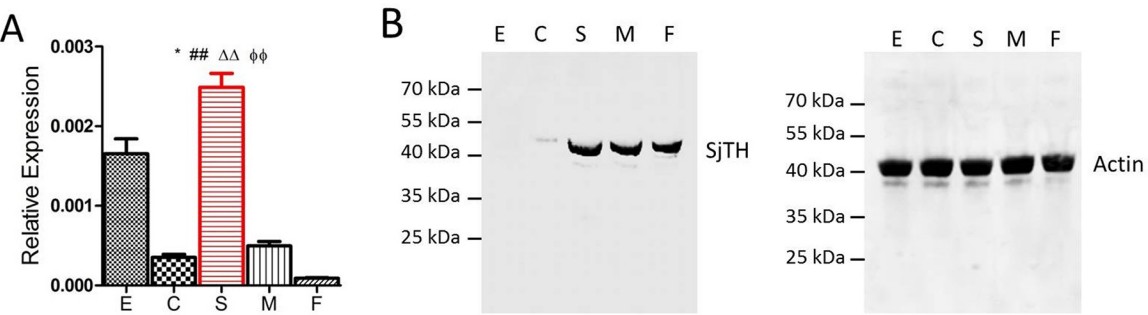

**Fig 1. Expression of SjTH in parasites at different developmental stages.** (A) Relative mRNA expression of *SjTH* in the five developmental stages of *S. japonicum*, cercariae (C), eggs (E), adult females (F), adult males (M, 42 days post-infection) and hepatic schistosomula (S, 14 days post-infection), was detected by qRT-PCR. Data represents the mean + standard deviation. *, #, Δ, and Φ indicate S group was compared with the E, C, M and F groups, respectively. *$p < 0.01$, ##, ΔΔ, ΦΦ$p < 0.01$ (Mann-Whitney test). (B) Expression of SjTH and Actin in the five developmental stages of *S. japonicum* detected by western blot.

schistosomiasis japonica with outstanding sensitivity and specificity [26], was used as positive control (Fig 3D). The recombinant SjTH exhibited promising sensitivity (100%, 95% confidence interval: 87.7–100%, two-tailed Student's *t*-test) and specificity (100%, 87.7–100%) for detecting *S. japonicum* infection. The results of SjSP-13 in this study were similar to those reported by Xu *et al.* (90.4% sensitivity and 98.9% specificity). Moreover, the antibody titers against SjTH were not significantly different between patients before and after 3-month praziquantel treatment.

## SjTH immunization protects against *S. japonicum* infection

To assess the immunoprotective role of SjTH against schistosome infection, BALB/c mice were immunized with His-tagged rSjTH, followed by challenge experiments. Immunization with rSjTH significantly prevented body weight loss in immunized and challenged mice (Fig 4A), but had no effect on splenomegaly and hepatomegaly in infected mice (Figs 4B and 4C and S2A). Next, each mouse was challenged with approximately 40 cercariae. The ratio of adult worms to cercariae in the control group was 71.0%, indicating that the cercariae were highly infective (Fig 4D). Immunization with rSjTH partially protected mice from *S. japonicum* infection, with a reduction in worm and egg numbers of 34.9% and 48.3%, respectively (Fig 4D and 4E), which led to decreased granulomatous area in the livers (Figs 4F and S2B), whereas SjSP-13 showed no protective effect. Furthermore, immunization with rSjTH significantly reduced the levels of interferon (IFN)-γ and tumor necrosis factor (TNF)-α (characteristic T helper type 1 [Th1] effector cytokines) in the serum of mice infected with schistosomes (Fig 5A and 5B), to levels close to those of uninfected mice. The characteristic T helper type 2 (Th2) effector cytokines, interleukin (IL) -4 and IL-10, were also reduced in the sera of rSjTH-immunized mice infected with schistosomes compared with control mice, but their levels were still higher than those of uninfected mice (Fig 5C and 5D).

## Discussion

Schistosomes rely on their neuromuscular systems to coordinate important locomotory behaviors, particularly the penetration of cercariae through the skin and subsequent migration of schistosomula in the bloodstream [7]. In addition to movement, the schistosome neuromuscular system also controls the muscles of the suckers, the muscle lining the reproductive, digestive, and excretory tracts, and the tight coupling of males and females [7]. These physiological functions are vital for the survival of these parasites. Thus, drugs that disrupt these

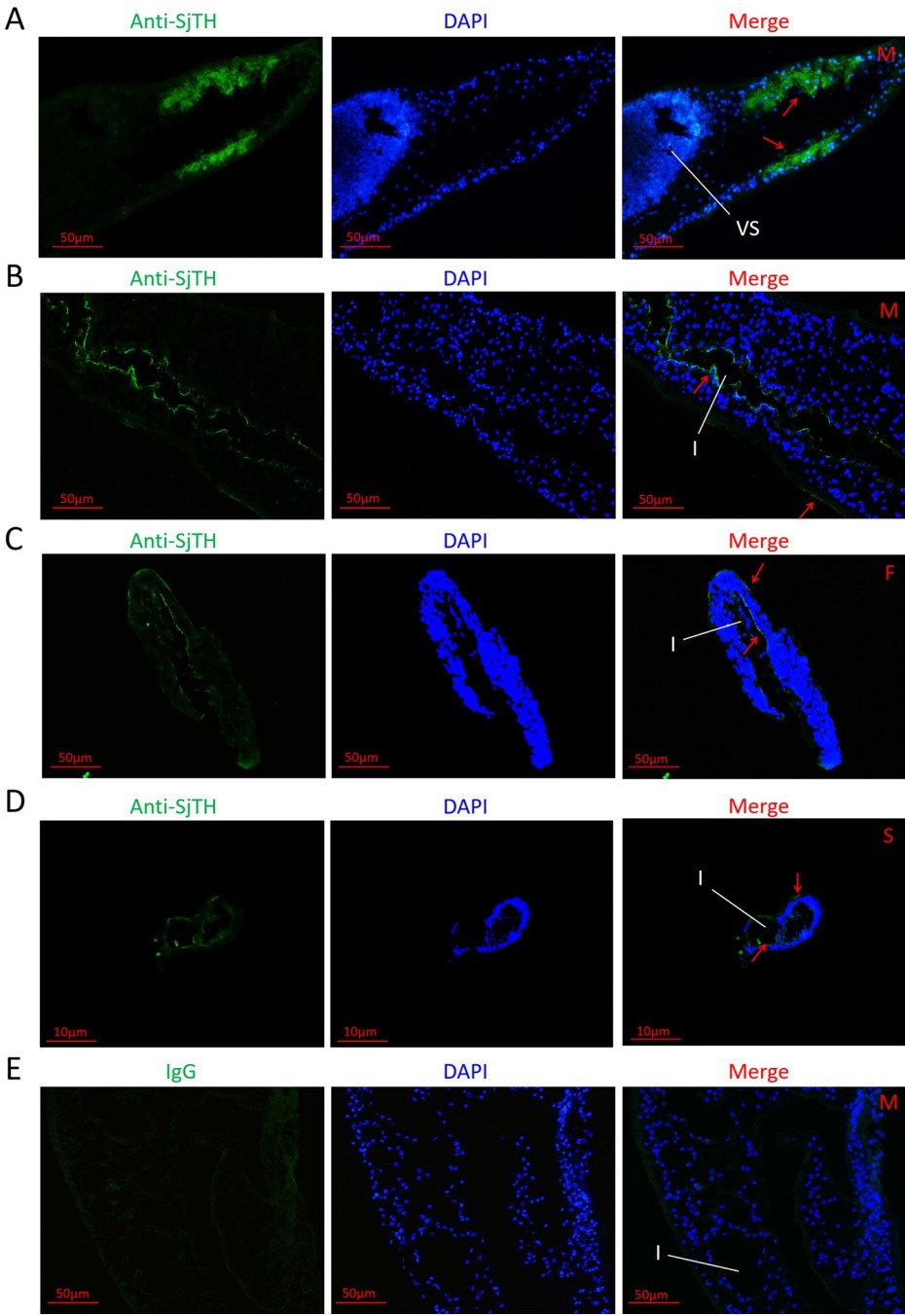

**Fig 2. Localization of SjTH in schistosome parasites.** Cryosections of anterior (A) of male adults, and mid-section of male adults (B, E), female adults (C), and hepatic schistosomula (D) were incubated with rabbit anti-SjTH polyclonal antibodies (A, B, C) or an IgG control (D), followed by Alexa Fluor 555 donkey anti-rabbit IgG (green fluorescence). The samples were counterstained with DAPI (in blue). Positive staining is indicated by red arrows. Results are representative of parasites from three independent experiments, and at least 3 worms of each experiment were analyzed for each stage. Abbreviations: F, female adult; M, male adult; S, hepatic schistosomula; I, intestine; VS, ventral sucker.

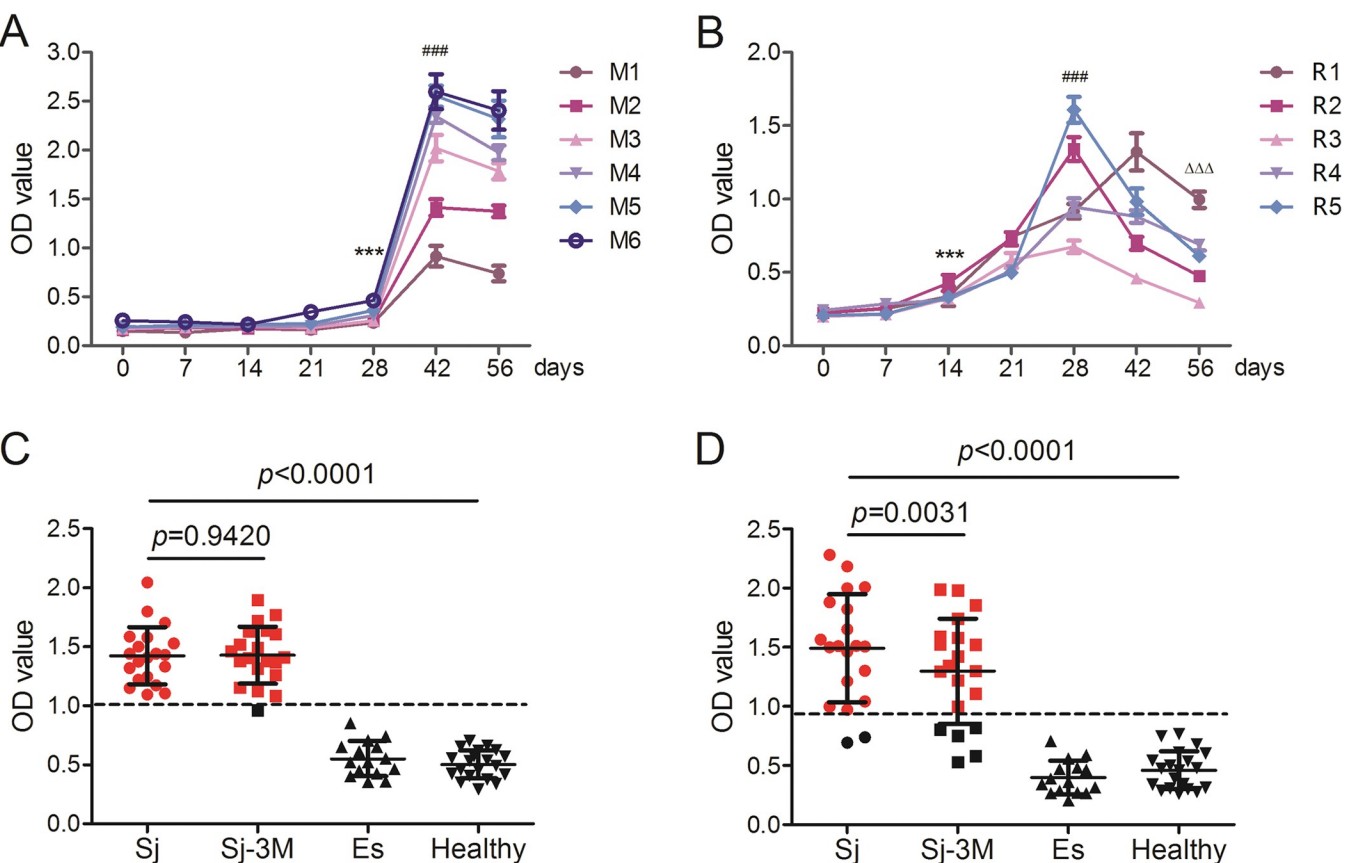

**Fig 3. Levels of SjTH-specific antibodies in host serum.** (A, B) BABL/c mice and New Zealand white rabbits were percutaneously infected with cercariae (40 ± 2 parasites per mouse and 1,000 ± 100 parasites per rabbit). Sera from infected animals were collected on days 0, 7, 14, 21, 28, 35, 42, and 56 post infection (dpi). Antibody titers against SjTH in these sera were determined by ELISA. The linear charts show the detailed dynamics of antibodies against SjTH. *, #, and Δ indicate comparisons with day 0, 28, and 42 dpi, respectively. ***, ###, ΔΔΔ indicate $p < 0.0001$. (C, D) SjTH-specific antibodies (C) and SjSP-13-specific antibodies (D) in human serum samples were determined by ELISA. Sera from patients with confirmed schistosomiasis japonica (Sj, $n = 20$), from the same patients 3 months after praziquantel treatment (Sj-3M, $n = 20$), from patients with confirmed echinococcosis (Es, $n = 15$), and from healthy individuals (healthy, $n = 20$) were included in the assay. The cutoff value for positive results was set at $\geq$ 2.1-times the mean optical density (OD) value of healthy individuals (dotted lines). Red and black dots indicate positive and negative individuals.

physiological functions are expected to interfere with the normal life cycle and ultimately eliminate parasites from the host. Praziquantel, the currently only available anti-schistosomal drug, works in part by disrupting the normal muscle function and causing paralysis in the worm [27,28]. Catecholamines cause muscular relaxation and lengthening of the parasite, thereby controlling movement [7–9]. TH is critical for normal physiological and neuropathological conditions as a rate-limiting enzyme in the synthesis of catecholamines [10,11]. Although schistosome TH has been discovered [20,21], little research has been done on its characteristics. Our results discovered SjTH not only distributed in the head associated with the CNS, but also expressed by the cells lining the membrane and the digestive tracts, which are involved in the movement, coupling and digestion of the parasites and associated with the PNS. Thus, it may be a promising target for anti-schistosomal therapy. However, the highly conserved catalytic domain of the enzyme makes it difficult to design drugs that can specifically act on the schistosome TH without affecting mammalian TH activity. Schistosome TH appears to be much closely related to mammalian enzymes, particularly bovine and human enzymes [20,21]. The greatest degree of sequence conservation among all species is in the C-terminal half to

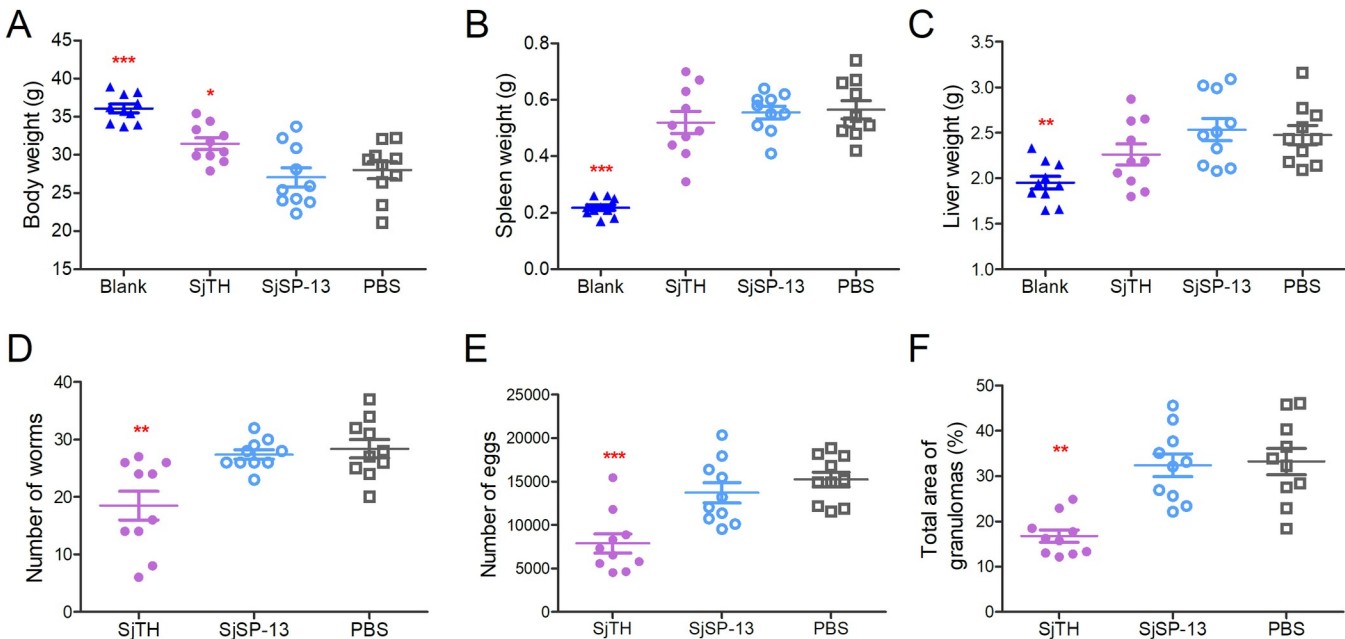

**Fig 4. Evaluation of the protective effect of SjTH in *S. japonicum*-infected mice.** To assess the protective role of SjTH as a vaccine candidate against schistosome infection, mice (*n* = 10 per group) were immunized with His-tagged recombinant SjTH, SjSP-13 (a novel diagnostic marker for schistosomiasis), or phosphate-buffered saline (PBS) as control. After immunization, the mice were challenged with cercariae (40 ± 2 per mouse). The blank group was immunized with PBS and received no cercariae. Results are representative of two independent experiments. The weight of the bodies (A), spleens (B) and livers (C) were determined and compared between the groups at 42 days post infection. The number of perfused adult worms (D) and liver eggs (E) was determined by microscopy at 42 days post infection. (F) Granuloma formation in livers was detected by haematoxylin-eosin staining. The proportion of total area of granulomas was compared between the groups. The results are representative of two independent experiments. Data are presented as mean±SD.* indicates $p < 0.05$, ** $p < 0.01$, and *** $p < 0.0001$ as compared with the control (PBS) group.

two-thirds of the TH, which comprises the catalytic domain of the enzyme [29–31]. SjTH has high homology with TH in mice (40%), humans (40%), and bovines (40%) [20].

In addition to drugs, vaccines represent the most cost-effective method for long-term control of schistosomiasis. Evidence from clinical and preclinical studies offers hope for developing an effective vaccine for long-term protection against schistosomiasis [32,33]. Over the past 2–3 decades, many candidate antigens have been characterized and tested against schistosome species. There are currently four schistosome antigens being evaluated in human clinical trials, including *S. haematobium* 28-kD glutathione S-transferase (rSh28GST) [34,35], *S. mansoni* 14-kDa fatty acid-binding protein (Sm14) [36], *S. mansoni* tetraspanin (Sm-TSP-2) [37], and the large subunit of *S. mansoni* calpain (Sm-p80) [38]. However, schistosome vaccines undergoing clinical trials are still lacking for schistosomiasis japonica, which is responsible for hepatic/intestinal disease (Asiatic or Oriental schistosomiasis) in the People's Republic of China, Philippines, and Indonesia [39]. Immunization with rSjTH reduced the worm and egg burden, and lessened the body weight loss in *S. japonicum*-challenged mice, which suggests that rSjTH may hold potential as a schistosomiasis vaccine candidate. However, the worm and egg reduction by rSjTH immunization remained limited. Compared to the 67% worm reduction of Sm14 [36] and 57% of Sm-TSP-2 [37] in immunized mice, as well as the 93% reduction of female adult worms by Sm-p80 in immunized baboons [38], the worm reduction of SjTH was only of 34.9%. Moreover, the 64% reduction in liver egg burden achieved by Sm-TSP-2 in immunized mice [37] and 90% reduction in tissue egg load by Sm-p80 [38] in immunized baboons were also much higher than the reduction in liver egg burden obtained with SjTH, which was of only 48.3%. Thus, further optimization of the rSjTH is warranted. Nevertheless,

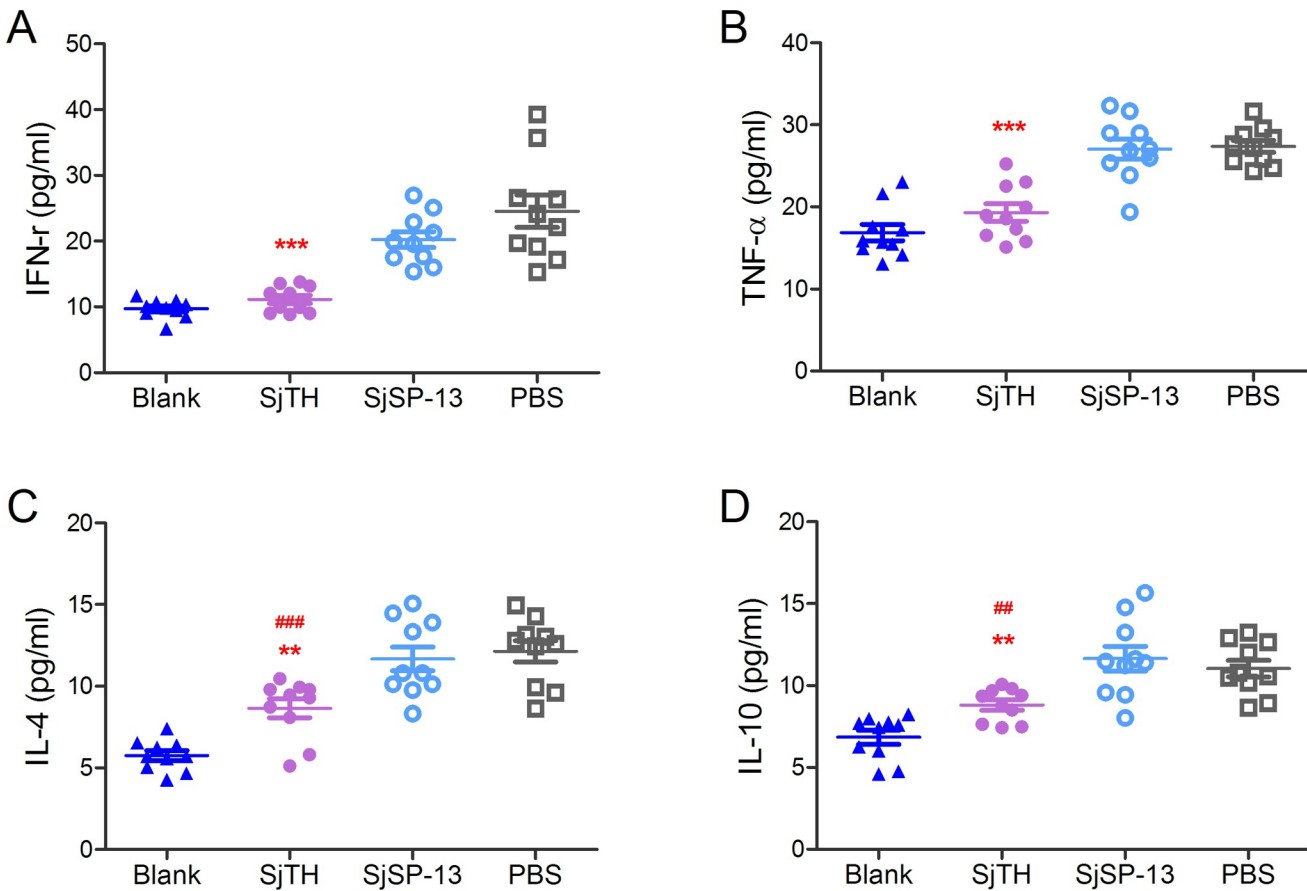

**Fig 5. SjTH immunization reduces cytokines of both Th1 and Th2 response in the sera of *S. japonicum*-infected mice.** Mice (*n* = 10 per group) were immunized with His-tagged recombinant SjTH, SjSP-13, or PBS as control, and were then challenged with cercariae (40 ± 2 per mouse). The blank group was immunized with PBS and received no cercariae. Results are representative of two independent experiments. The levels of interferon (IFN)-γ (A), tumor necrosis factor (TNF)-α (B) interleukin (IL)-4 (C) and IL-10 (D) in the sera 42 days post infection were determined by ELISA and compared between the groups. # and * indicate comparisons with the blank and control (PBS) groups, respectively. * indicates $p < 0.05$, **, ## $p < 0.01$, and *** $p < 0.0001$.

is noteworthy that the effectiveness of *S. japonicum* antigens is generally lower than that of *S. mansoni* antigens [39]. *S. japonicum* antigens currently under research mainly include *S. japonicum* paramyosin (Sj97), *S. japonicum* triosephosphate isomerase (SjTPI), *S. japonicum* cytosolic fatty acid-binding protein (SjFABPc), *S. japonicum* 23-kDa integral membrane protein (Sj23), and *S. japonicum* 16-kDa surface protein (Sj16) [40–43]. Worm burden reduction with Sj97 [40], SjTPI [41], SjFABPc [42], and Sj23 [43] in mice reached 32%, 31.9%, 49%, and 45.5%, respectively; and the reduction in liver egg burden was of 34–66%, 13.7–18.6%, 27.2%, 58.4%, respectively. Our recent study reported a group of novel hepatic schistosomula antigens as potential schistosomiasis japonica vaccine candidates, with the most promising antigen SjScP25 generating about 50% worm reduction and 65% egg reduction in mice challenged with infective cercariae [44]. Thus, for *S. japonicum* infection, the discovery of additional candidate antigens cannot be overemphasized. In addition, compared with single-antigen immunization, multiantigen-combined immunization may be a more feasible direction of vaccine development for schistosomiasis, in particular for schistosomiasis japonica.

About 5–6 weeks post infection, schistosome induces Th1 dominant responses of the host with the upregulation of effector cytokines (such as TNF-α and IFN-γ) [44,45]. Soluble egg antigens (SEA) are key factors driving the dramatic transition from a Th1 to a Th2 dominant

response. IL-4 is recognized as the dominant cytokine for Th2 response and granuloma development, as the Th1 response during the early stages of schistosomiasis is downregulated by IL-4 and -10 [46,47]. Murine studies have shown that extreme immune deviation toward either Th1 or Th2 response induced by schistosome results in increased pathology and premature death [47,48]. Therefore, maintaining Th1/Th2 balance is important for controlling the excessive pathology of schistosomiasis [49]. In this study, both serum Th1 and Th2 cytokine levels in rSjTH immunized mice decreased with the decrease of the number of worms and eggs. Whether rSjTH immunization may contribute to maintain systemic Th1/Th2 response balance in schistosomiasis remains to be further investigated.

Although SEA is the most widely used antigen for the diagnosis of schistosomiasis, its preparation cost is high and its yield is low, while recombinant antigen is low in cost and easier to prepare, so it is more suitable for schistosomiasis immunodiagnosis [50]. In recent years, the immunodiagnostic techniques of schistosomiasis japonica have developed rapidly, many novel schistosome antigens for enzyme-linked immunosorbent assay with both high sensitivity and specificity and has been discovered, such as, *S. japonicum* recombinant phosphoglycerate mutase (rSjPGM) [51], *S. japonicum* Saposin-like protein 4 (SjSAPLP4) and SjSAPLP5 [25], *S. japonicum* cathepsin B (SjCatB) [52], SjScP25 [44]. In this study, SjTH also showed high antigenicity and effectively stimulated the humoral immune response in mammalian hosts, and the herein described rSjTH, the nonconserved fragment of naïve SjTH, exhibited excellent sensitivity and specificity for the diagnosis of *S. japonicum*.

In summary, our study revealed that SjTH, which is mainly expressed in the intestinal tract and membranes of larvae and adult worms of *S. japonicum*, not only exhibits promising sensitivity and specificity for schistosomiasis japonica immunodiagnosis, but also holds potential value as an immunoprotection target. Hence, SjTH may be helpful to control *S. japonica* infection and spreading. However, this study about SjTH is still in the preliminary stage, the exact location of SjTH and its specific mechanism in schistosome development need to be further investigated.

## Supporting information

**S1 Table. Clinical characteristics of the enrolled subjects whose sera were used in ELISA for diagnosis.**
(DOCX)

**S1 Fig. Detection of recombinant protein of SjTH by SDS-PAGE and western blot.** Recombinant protein of SjTH (71–181 amino acids) were resolved in 12% SDS-PAGE and stained by Coomassie brilliant blue staining (Lane 1), and then detected by western blot with an anti-His tag mouse monoclonal antibody (Lane 2), a mixture of serum samples (equal volumes) from 10 schistosomiasis japonica patients (Lane 3) or 10 healthy volunteers (lane 4), a mixture of serum samples (equal volumes) from 6 infected BALB/c mice 42 days p.i. (Lane 5) or normal mice (Lane 6), and a mixture of serum samples (equal volumes) from 5 infected rabbits 42 days p.i. (Lane 7) or normal rabbits (Lane 8).
(TIF)

**S2 Fig. Pathological changes of livers and spleens in SjTH immunized mice.** Mice (*n* = 10 per group) were immunized with His-tagged recombinant SjTH, SjSP-13, or PBS as control, and were then challenged with cercariae (40 ± 2 per mouse). The blank group was immunized with PBS and received no cercariae. Results are representative of two independent experiments. (A) The morphology of the livers and spleen from each group of mice are shown. (B)

Granuloma formation was detected by haematoxylin-eosin staining.
(TIF)

## Acknowledgments

We thank Jiangxi Provincial Institute of Parasitic Diseases for providing *S. japonicum* infected Snails. We thank editage ([www.editage.cn](www.editage.cn)) for language polishing.

## Author Contributions

**Conceptualization:** Nan Hou.

**Data curation:** Xianyu Piao, Jiamei Duan.

**Formal analysis:** Xianyu Piao, Ning Jiang, Nan Hou.

**Funding acquisition:** Nan Hou, Qijun Chen.

**Investigation:** Xianyu Piao, Jiamei Duan, Shuai Liu, Nan Hou.

**Methodology:** Xianyu Piao, Nan Hou.

**Project administration:** Nan Hou.

**Resources:** Ning Jiang, Nan Hou, Qijun Chen.

**Software:** Jiamei Duan.

**Supervision:** Nan Hou, Qijun Chen.

**Validation:** Xianyu Piao, Jiamei Duan.

**Visualization:** Xianyu Piao, Nan Hou.

**Writing – original draft:** Nan Hou.

**Writing – review & editing:** Ning Jiang, Nan Hou, Qijun Chen.

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
