## [Decision Letter · Decision Letter 0]

26 Jan 2023

Dear Dr. Hou,

Thank you very much for submitting your manuscript "Schistosoma japonicum Tyrosine Hydroxylase is Promising Targets for Immunodiagnosis and Immunoprotection of Schistosomiasis japonica" for consideration at PLOS Neglected Tropical Diseases. As with all papers reviewed by the journal, your manuscript was reviewed by members of the editorial board and by several independent reviewers. In light of the reviews (below this email), we would like to invite the resubmission of a significantly-revised version that takes into account the reviewers' comments. 

We cannot make any decision about publication until we have seen the revised manuscript and your response to the reviewers' comments. Your revised manuscript is also likely to be sent to reviewers for further evaluation.

Sincerely,

Michael H. Hsieh

Academic Editor

Eva Clark

Section Editor

Reviewer's Responses to Questions

**Key Review Criteria Required for Acceptance?**

**Methods**

-Are the objectives of the study clearly articulated with a clear testable hypothesis stated?

-Is the study design appropriate to address the stated objectives?

-Is the population clearly described and appropriate for the hypothesis being tested?

-Is the sample size sufficient to ensure adequate power to address the hypothesis being tested?

-Were correct statistical analysis used to support conclusions?

-Are there concerns about ethical or regulatory requirements being met?

Reviewer #1: (No Response)

Reviewer #2: See general comments

Reviewer #3: The objectives are clearly stated. The study design is appropriate. The population is described- there are some comments in my narrative on the possible effect of the egg counts among the human participants. This is an experimental, explorative study, which requires further investigation for validation, so the sampling is ok in this context. The ethics and regulations seem to be adhered to. I am however concerned about a similar abstract from a paper published in 2022 with some of the same authors- I cannot access the full text- but would like this to be checked by the editor.

**Results**

-Does the analysis presented match the analysis plan?

-Are the results clearly and completely presented?

-Are the figures (Tables, Images) of sufficient quality for clarity?

Reviewer #1: (No Response)

Reviewer #2: See general comments

Reviewer #3: The results are displayed as per the methodology. There is a comment on Table 1. and Figure 1 as per my narrative for the authors consideration.

**Conclusions**

-Are the conclusions supported by the data presented?

-Are the limitations of analysis clearly described?

-Do the authors discuss how these data can be helpful to advance our understanding of the topic under study?

-Is public health relevance addressed?

Reviewer #1: (No Response)

Reviewer #2: See general comments

Reviewer #3: The authors clearly state that there is limited research in this area, however they need to consider including some updated references in their background- I have made a suggestion of a recent review paper that they could look at. The authors also need to include some background on the burden of disease relating specifically to S.japonicum. 

The authors need to consider strengthening their statement in lines 311 and 312 about the “strong evidence” supporting vaccine development and how this relates to the present findings.

**Editorial and Data Presentation Modifications?**

Reviewer #1: (No Response)

Reviewer #2: See general comments

Reviewer #3: the authors need to check how they write S japonicum- this needs to be italicized in some instances the scientific way of writing this must be adhered to.

**Summary and General Comments**

Reviewer #1: (No Response)

Reviewer #2: This study Piao et al., (PNTD-D-22-01627) characterizes Schistosoma japonicum Tyrosine hydroxylase (TH), an important catecholamine biosynthesis enzyme, in its tissue expression and its roles in protection against infection and host immune response. Given its high similarity to the human counterpart and the minimal protective effects reported, SjTH does not appear to have a superior translational value. Moreover, the experimental results do not fully support the claims made in the manuscript. However, this is an important molecule/pathway that can shed light on the biology of schistosomes, especially regarding its role in neuromuscular function. In the reviewer’s opinion, there are several major and minor aspects of that can be addressed to improve the quality of the manuscript. 

Major Comments: 

1) Questionable TH protein expression pattern. Previous work that are cited in the manuscript describe neuronal enrichment of catecholamines. For instance, Gianutsos et al., 1977 demonstrates that the highest enrichment of dopamine and norepinephrine is seen in the anterior region where the cephalic ganglia (central nervous system) are located. This is further supported by the fact Smp_007690 (S. mansoni ortholog of SjTH) mRNA is enriched in a subset of neurons in adult worms (Wendt et al., 2021). Together with reported effects of catecholamines in circular/longitudinal muscle contraction (Pax et al., 1984), these data support a model in which TH enzyme in the CNS produces catecholamines that leads to proper functioning of peripheral neuromuscular function. Thus, the antibody staining results that show protein expression in the gut lumen is questionable. Expression pattern in the CNS should be shown as a comparison, and if there is no enrichment seen in the CNS, the authors need to explain the discrepancy between the predicted and observed expression patterns. If mRNA in situ hybridization cannot be done, alternatively, the authors can show the PCA plot of the single cell RNA-seq atlas available for S. mansoni (see examples shown in Hulme et al., 2022 Supplementary Figures) and discuss the biological/technical reasons for the discrepancy. 

2) Lack of mechanistic details. Although the study aims to understand the “in vivo” role of TH, and the apparent moderate protective role shown, the authors come short on investigating the reasons behind the reduced parasite burden. Are parasites unable to migrate from the skin to the bloodstream? Are parasites’ feeding, reproductive, or locomotive behavior affected? Harvesting parasites at different times after infection and comparing the recovery efficiency would provide some answers. In parallel, monitoring the behavior of harvested parasites would give some clues. Another important mechanistic detail is that if the antibody against rSjTH indeed targets and neutralizes SjTH, one might expect to see reduced levels of catecholamine. These should nicely corroborate and support reduced parasite/egg burden results. 

Minor Comments: 

1) Information regarding the Gene ID should be specified in the Methods: Sjp_0053450

2) Figure 1 has a discrepancy in expression between SjTH mRNA and protein (especially in eggs where mRNA is highly expressed but the protein is not detected). The authors need to explain why there is such a big discrepancy in the text. 

3) Figure 1. Why is y-axis of qPCR data shown in 0.001 units? Please describe step-by-step analysis on how the values were derived in the Methods section or in the Figure Legend. 

4) In Figure 1, ‘hepatic schistosomula’ stage has not been defined. Indicate how many days post-infection the worms were harvested for the reported experiments.

5) Figure 2. The image qualities are poor to really see the signal. In fact, all figures are severely pixelated. 

6) Figure 2. How many worms were analyzed for each stage, and how many of those showed such expression patterns need to be indicated. 

7) Figure 2. In panels in D, DAPI and IgG panels appear to be swapped. 

8) Figure 2. The scale bar units should be in µm (micrometer), not µM (micromolar). 

9) Why was His-rSjTH used for immunization instead of epitope-cleaved rSjTH? Since His tag was not removed, in this case, a better negative control would be an epitope-only immunization. 

10) Figure 4. Representative liver (highlighting the reduced pathology of granulomatous liver) and spleen images will nicely complement Figures 4B and 4C. 

11) Figure 4. The authors should add a comment regarding the sex of schistosomes infected/recovered. From how I understood the paper, the authors used mixed sex cercariae derived from infected Oncomelania. Given the relatively small number (~40) of cercariae given to each mouse, a small variation in male:female ratio could shift the egg output and/or Th1/Th2 response. This needs to be discussed in the text. 

12) Figure 5. The conclusion that SjTH may modulate Th1/Th2 response is overreaching. Given that rSjTH immunization results in a moderate reduction of parasite burden (shown in Figure 4), the overall reduction in immune response could simply be due to the lower number of parasites and eggs in rSjTH infected mice. This needs to be clarified in the text. 

13) Many references are missing full citation, and are not properly formatted (e.g., italicized species name).

Reviewer #3: The findings have merit and according to the authors there is a scarcity of knowledge in this area. My concern is around the similarity of a possible dual publication- which I cannot access the full text for. the authors need to explain how this is different to the present paper.

PLOS authors have the option to publish the peer review history of their article (what does this mean?). If published, this will include your full peer review and any attached files.

Reviewer #1: No

Reviewer #2: No

Reviewer #3: No
---

## [Decision Letter · Decision Letter 1]

18 May 2023

Dear Dr. Hou,

We are pleased to inform you that your manuscript 'Schistosoma japonicum Tyrosine Hydroxylase is Promising Targets for Immunodiagnosis and Immunoprotection of Schistosomiasis japonica' has been provisionally accepted for publication in PLOS Neglected Tropical Diseases.

Best regards,

Michael H. Hsieh

Academic Editor

Eva Clark

Section Editor

Reviewer's Responses to Questions

**Key Review Criteria Required for Acceptance?**

**Methods**

-Are the objectives of the study clearly articulated with a clear testable hypothesis stated?

-Is the study design appropriate to address the stated objectives?

-Is the population clearly described and appropriate for the hypothesis being tested?

-Is the sample size sufficient to ensure adequate power to address the hypothesis being tested?

-Were correct statistical analysis used to support conclusions?

-Are there concerns about ethical or regulatory requirements being met?

Reviewer #2: See General Comments

Reviewer #4: All points mentioned above regarding the methods are well designed and accepted.

**Results**

-Does the analysis presented match the analysis plan?

-Are the results clearly and completely presented?

-Are the figures (Tables, Images) of sufficient quality for clarity?

Reviewer #2: See General Comments

Reviewer #4: The results are well presented and matching the plan

**Conclusions**

-Are the conclusions supported by the data presented?

-Are the limitations of analysis clearly described?

-Do the authors discuss how these data can be helpful to advance our understanding of the topic under study?

-Is public health relevance addressed?

Reviewer #2: See General Comments

Reviewer #4: The conclusions are fine and supporting the the methods and results.

**Editorial and Data Presentation Modifications?**

Reviewer #2: See General Comments

Reviewer #4: (No Response)

**Summary and General Comments**

Reviewer #2: (No Response)

Reviewer #4: The article is well written, the content fulfill the aim and the discussion is logic.

PLOS authors have the option to publish the peer review history of their article (what does this mean?). If published, this will include your full peer review and any attached files.

Reviewer #2: No

Reviewer #4: No

---

## [Editor Report · Acceptance letter]

26 May 2023

Dear Dr. Hou,

We are delighted to inform you that your manuscript, "Schistosoma japonicum Tyrosine Hydroxylase is Promising Targets for Immunodiagnosis and Immunoprotection of Schistosomiasis japonica," has been formally accepted for publication in PLOS Neglected Tropical Diseases.

Best regards,

Shaden Kamhawi

co-Editor-in-Chief

Paul Brindley

co-Editor-in-Chief
